# Therapeutic Effects of Intradialytic Exercise on Life Quality of Patients with End-Stage Renal Disease: Study Protocol for a Randomized Control Trial

**DOI:** 10.3390/healthcare10061103

**Published:** 2022-06-14

**Authors:** Hsiang-Chi Chang, Cheng-Hsu Chen, Yuan-Yang Cheng

**Affiliations:** 1Department of Physical Medicine and Rehabilitation, Taichung Veterans General Hospital, Taichung 407219, Taiwan; s19801041@gm.ym.edu.tw; 2Department of Post-Baccalaureate Medicine, College of Medicine, National Chung Hsing University, Taichung 402010, Taiwan; cschen920@yahoo.com; 3Department of Nephrology, Taichung Veterans General Hospital, Taichung 407219, Taiwan; 4Department of Life Science, Tunghai University, Taichung 407224, Taiwan; 5School of Medicine, National Yang Ming Chiao Tung University, Taipei 112304, Taiwan

**Keywords:** depression, exercise therapy, hemodialysis, physical fitness, quality of life

## Abstract

Background: Exercise training has positive effects on physical functions and could reduce a sedentary lifestyle for hemodialysis (HD) patients. Given that low-level physical activity increases morbidity and mortality, here, we aimed to determine the effects of an intradialytic exercise program delivered at different frequencies on HD patients in Taiwan. Methods: This study is a prospective, randomized control trial. An intradialytic exercise program will be arranged for patients after receiving their informed consent. Patients will be segregated at random into three groups as follows: (a) three times/week of intradialytic exercise training plus standard care maintenance of HD, (b) two times/week of intradialytic exercise training plus standard care maintenance of HD, or (c) standard care maintenance of HD. Subjects will be followed for 24 weeks. At three time points, 0, 12, and 24 weeks, the primary outcome, the Short-Form 36 score, will be measured. Additional secondary outcomes to be measured are the Beck depression inventory, 6 min walking test, sit-to-stand test, and anthropometric measures such as the body mass index, thigh circumference, and the proportion of fat in the body composition. Conclusions: There is emerging evidence in support of intradialytic exercise improving health-related quality of life for patients on HD. However, the difference in the therapeutic effects between three times per week and twice per week has never been determined. With this study, we anticipate to fill the knowledge gap in the exercise prescription among HD patients.

## 1. Introduction

Patients with chronic kidney disease (CKD) constitute a large portion of the population in modern society. They cause a tremendous amount of medical expenditure. In the United States, ≥30 million adults (~14.8% of the adult population) have CKD [1]. These patients rely on either maintenance dialysis or a renal transplant, as CKD progresses to end-stage renal disease (ESRD). In Taiwan, the prevalence of hemodialysis (HD) ranks top in the world; 15–30% higher in the HD rate compared with the United States. In Taiwan, the national prevalence of hemodialysis is 3587 per million people, and ~94,000 patients undergo HD every year. The number of HD increases by 3% to 4% annually [2]. HD is a lengthy process repeating two to three times per week, with each session lasting 3 to 4 h.

There is good evidence that a low level of physical activity is associated with increased morbidity and mortality [3,4,5,6]. Those patients classified as stage 5 CKD typically have a low functional capacity, which is about only 50–80% of their sex- and age-matched controls [7]. Patients on HD have higher rates of falls, hospitalization, morbidity, and mortality [8]. Exercise training in CKD patients can lead to a blood pressure reduction and improvement in aerobic capacity, heart rate variability, muscular function, and health-related quality of life (HRQoL) [9]. Given the benefits of exercise training, the American College of Sports Medicine (ACSM) guidelines recommend that CKD patients, including those with ESRD, should gradually progress from low-intensity exercise of 10–15 min in duration to a greater volume of exercise [10]. Intradialytic exercise is engaged with the use of pedaling and stepping devices, while a patient is seated in a dialysis chair. In the United Kingdom (UK), structured exercise interventions, mostly delivered by a cycle ergometer during HD at medical centers, have shown clinical benefits in patients. These were also shown to improve HD adequacy, have anti-depressive effects, cost-effective outcomes, and improve quality of life in these British patients [11,12].

Despite the benefits aforementioned, the intradialytic exercise program is not commonly incorporated into regular HD care worldwide. After all, the formal recommendation from the ACSM is to perform exercise training for a minimum of two hours after, but not during, HD [10]. In Taiwan, no HD unit incorporates an exercise program as part of regular care. Very few studies have evaluated the effects of structured intradialytic exercise programs on the level of physical activity and HRQoL of HD patients [13,14,15]. According to the ACSM’s guideline for an exercise prescription [10], the frequency of aerobic exercise for HD patients is 3 to 5 days per week. Whether a low frequency of exercise (such as two days per week) produces similar therapeutic effects is unclear. The objective of this randomized control trial is, therefore, to compare the efficacy of different frequencies of structured intradialytic exercise programs on Asian HD patients in Taiwan.

## 2. Materials and Methods

### 2.1. Study Design Overview

This study is a randomized control trial in which participants will be grouped into 3 regular maintenance dialysis groups, with intradialytic exercise given (a) twice per week and (b) three times per week. The outcome measures will be measured at 3 time points: (a) right after the recruitment of participants, and after the initiation of the study, (b) 3 months later, and (c) 6 months later. The timings of the outcome measurements will be set according to past studies on intradialytic exercise. Some of them measured the results after 3 months of intradialytic exercise [13,14,15], while others were evaluated 6 months later [11,16].

### 2.2. Study Participants

#### 2.2.1. Selection Criteria

Patients receiving HD in a tertiary medical center in Taiwan will be screened for eligibility. Participants must be ≥18 years old and have received HD treatments without adverse events for ≥90 days. Considering that different baseline exercise habits may pose a significant confounding factor in this study, having no previous regular exercise habits is also listed to be one of our inclusion criteria. The exclusion criteria include: patients with an expected survival of ≤6 months, patients being withdrawn from HD within 6 months (such as those likely to receive a kidney transplantation or transfer to peritoneal dialysis), a Charlson comorbidity index (CCI) of >10, lower-limb amputations, or conditions such as dementia and/or psychiatric disorders that would not allow patients to complete the outcome measures of this study.

#### 2.2.2. Randomization

After signing informed consent, participants will be randomly divided into three groups: (i) 3 times/week of intradialytic exercise training plus usual care maintenance of HD, (ii) 2 times/week of intradialytic exercise training plus usual care maintenance of HD, and (iii) usual care maintenance of HD. Randomization of the allocation sequence will be completed using computer-generated random numbers, while taking age, gender, and diabetes status into consideration. A project assistant will examine the outcome measures, and both the participants and the assistant will not be blindly involved in this study.

### 2.3. Intervention

#### 2.3.1. Usual Care

All participants will receive HD as prescribed and will be adjusted by attending nephrologists. The HD routine care includes the management of blood pressure, the treatment of anemia, the control of electrolyte level, and mitigation strategies of cardiovascular risk. All participants will receive HD using the Fresenius 4008S machine. The participants will be encouraged to maintain their regular physical activities as usual, and no extra exercise training should commence during the 6 months of the follow-up period.

#### 2.3.2. Intradialytic Exercise

The intradialytic exercise program is based on the leading CYCLE-HD program from a study group in the United Kingdom [16]. Participants perform 30 min of continuous cycling at a 12 to 13 rating of perceived exertion (RPE) during HD for a total of 6 months. The cycling exercise is executed during the first 2 h of HD with participants lying in supine posture. Specially adapted and calibrated cycle ergometers (WINGS PLUS WP-948) are used for exercise training. Participants in the intradialytic exercise groups will receive cycling training either 2 or 3 times a week during HD. Resistance training for bilateral knee extensors using ankle weights will be incorporated after the aerobic cycling exercise for the participants. Exercise duration, intensity (RPE), blood pressure, and heart rate will be monitored for each exercise session. As in the usual care group, these participants will also be encouraged to keep their original daily activities without additional exercise training during the study period.

### 2.4. Data Collection and Outcomes

Data will be collected according to the study timeline as displayed in Figure 1. Clinical data, which include blood pressure, heart rate, and body weight, will be obtained before and after the routine care protocols of HD from electronic health records. Adverse events such as hypotensive episodes, cramping, nausea, vomiting, or loss of consciousness will be monitored by clinical personnel. Results of the baseline laboratory tests (e.g. electrolytes, albumin, complete blood count, and dialysis efficiency using Kt/V) will also be extracted from the electronic health records.

#### 2.4.1. Primary Outcomes

Our primary outcome measure of patients is the Short-Form 36 (SF-36) questionnaire, which evaluates the health-related quality of life of the participants. The validity and reliability of the Chinese version of this questionnaire were confirmed in a previous study [17]. Patients who are illiterate or have visual problems will be provided verbal assistance when completing the questionnaire. The SF-36 questionnaire includes 8 subdomains, namely, vitality, physical function, bodily pain, general health perception, physical role functioning, emotional role functioning, social role functioning, and mental health. The lower the score, the lower the self-rated quality of life. These 8 subdomains are integrated into two domains: the physical component scale (PCS) and the mental component scale (MCS).

#### 2.4.2. Secondary Outcomes

Secondary outcome measures include the Beck depression inventory (BDI) to evaluate the status of depression, 6 min walking test (6MWT) to determine cardiopulmonary endurance, sit-to-stand test (STS-30) to measure lower-limb muscle endurance, and anthropometric measures, such as body mass index (BMI), thigh circumference, and the fat proportion in the body composition as obtained from the bioimpedance analysis (Tanita MC-780).

A 21-item, self-rated BDI with good reported validity and reliability will be applied to measure the depression status of the participants [18,19]. This BDI includes symptoms of depression such as hopelessness and irritability, cognitions such as guilt or feelings of being punished, as well as physical symptoms such as fatigue, weight loss, and lack of interest in sex. Subjects will rate each question on a 4-point scale. The severity of depression is represented by the summed score that ranges from 0 to 63. Scores of 0–13 indicate minimal depression, 14–19 indicate mild depression, 20–28 indicate moderate depression, and 29–63 indicate severe or major depression.

Physical function such as cardiopulmonary endurance will be assessed with 6MWT, and the muscle endurance of lower limbs will be assessed with STS-30. 6MWT is a self-paced test of walking capacity to reflect the functional ability of daily activities [20]. Participants will walk along a walkway (30 m long, 2 m wide) in a quiet corridor of the hospital. Markers will be placed on the walkway at 5 m intervals. Participants will be instructed to walk as far as they can within a period of 6 min, but are allowed to stop and rest if necessary. STS-30 is an indirect measure of lower-limb muscle strength and endurance [21]. Using a chair without arms placed against the wall to maintain stability, the seat height will be adjusted to allow the thigh to be parallel to the ground. The STS-30 test measures the repeated number of times a patient can stand up and sit down from a chair repeatedly during a 30 s period while their arms are crossed in front of their chest.

### 2.5. Statistical Analysis and Power Estimates

Statistical analyses will be performed using SPSS, version 22.0 (SPSS Inc., Chicago, IL, USA). The Shapiro–Wilk test of normality will be used to evaluate the distribution of each outcome variable. If data are normally distributed, paired t*tests and analysis of variance (ANOVA) will be used. If data are not normally distributed, the Wilcoxon signed-rank test and Kruskal–Wallis test will be used. The preintervention outcome measures will be compared using ANOVA or Kruskal–Wallis test. Paired t-tests or Wilcoxon signed-rank test will be applied to compare differences between pre- and post-tests. After the exercise intervention, intergroup comparisons regarding improvements of outcome measures will be determined using ANOVA or Kruskal–Wallis test.

The G*Power (Germany, version 3.1.9) software will be used for sample size estimation [22]. Based on repeated measures ANOVA, a statistical power of 0.8, a significant level of 0.05, the effect size is 0.35, refs. [23,24,25], and a minimum of 84 participants are required for our study. We noted the likelihood of a high drop-out rate for intradialytic exercise [26].

### 2.6. Safety

The criteria for immediate interruption of the exercise includes intense physical exhaustion, chest pain, dyspnea, dizziness, unstable heartbeat and blood pressure, cramping, or fatigue of the lower limbs. For each participant, serious adverse events and reasons for withdrawal from this study will be recorded accordingly.

## 3. Discussion

Despite good evidence on the benefit of exercise training on physical function, and that such training could reduce a sedentary lifestyle among HD patients, the benefits related to different frequencies of intradialytic exercise protocols in Asian populations is largely unclear. While an intervention with intradialytic exercise is generally considered safe among diverse racial and cultural populations, the best frequency of interventions remains to be investigated.

HD is associated with a high prevalence of psychological problems [27]. Among the HD population, depression is the most common mental disorder. Furthermore, the prevalence of depression in these patients is higher than those in other chronic disease populations [27]. A previous study reported that, based on the mental component scale of SF-36, Taiwanese HD participants had scores significantly lower than the general population [28]. While a recent systematic review advocated that regular exercise may reduce depression and fatigue in HD patients [29], results are still inconsistent across studies [30,31,32]. Hence, our study aims to provide useful information bridging the knowledge gap in Asian populations.

Current evidence supports that moderate-intensity exercise training effectively improves HRQoL [33]. In addition, two systematic reviews and meta-analyses showed that intradialytic exercise is associated with better HRQoL [23,31]. However, another two studies reported no significant difference in HRQoL after intradialytic exercise [29,34]. Regarding the mode of intradialytic exercise, Gomes et al. reported that aerobic exercise alone is not significantly associated with HRQL [32]. In our proposed study on patients under HD, we will incorporate resistance training for knee extensors after the aerobic cycling exercise. Together, they will likely constitute a more comprehensive exercise program.

Systematic reviews and meta-analyses found a significantly improved PCS in the intradialytic exercise group, but not in the MCS [32,35,36]. However, some subdomains of the MCS are still improved after intradialytic exercise. These subdomains include “limitations due to emotional problems” scores in an Iranian study [37], and “bodily pain” scores in a Korean study [38]. As mental problems constitute an important component of life quality, they should not be overlooked in HD patients. Compared with the US general population and US HD cohorts, Taiwanese HD patients have substantially lower physical and mental aspects of life quality [29]. It is well known that for HD patients, their poor physical dimension of HRQoL predicts higher mortality [39]. Therefore, our study will use the SF-36 score as our primary outcome measure, aiming to evaluate the effects of different frequencies of intradialytic exercise on HRQoL.

Blood pressure is regulated through various neural mechanisms involving the sympathetic nervous system, vasoconstriction, and baroreflex activity. During HD, a patient may experience acute hypotension, primarily due to hypovolemia and poor compensatory mechanisms to counter the reduced intravascular volume. In addition, autonomic dysfunction is highly prevalent in these patients, and it has been implicated in the increased risk of blood pressure instability during HD [40,41,42]. In the supine position, SBP is lower due to the reduced effect of gravity on blood in its return to the heart [43]. HD is typically conducted in either a lying or a seated position. While exercise-induced hypotension is of concern for patients and their healthcare givers, our study protocol involves intradialytic exercise in a supine posture, simply to minimize the exercise-induced hypotension.

Most intradialytic exercises in the literature, including the best-known CYCLE-HD study, were conducted for 30 min and given three times a week [11,13,14,15,16]. There is only one study that used the exercise frequency of two times a week, and the training mode involved resistance training and stretching, but not aerobic training [44]. Some other studies also adopted the exercise frequency of two to three times a week, such as those of Vilstreren et al., Chigira et al., and Bae et al. [45,46,47]. To our knowledge, no study has yet directly compared intradialytic exercise groups specifically between three times and two times a week. Since cycle ergometer devices and healthcare givers monitoring intradialytic exercise may be limited in some hospitals in different countries, whether the frequency of intradialytic exercise could be reduced is a crucial and practical issue that needs to be clarified. Hence, we have designed this comparative study in order to clarify this issue and try to fill the knowledge gap in the field of intradialytic exercise.

## 4. Conclusions

In summary, our protocol will evaluate the therapeutic effects of intradialytic exercise programs for HD patients given at different frequencies. With our results, we should be able to determine whether lowering the frequency of intradialytic exercise brings similar therapeutic effects in the Asian population of Taiwan.

## Figures and Tables

**Figure 1 healthcare-10-01103-f001:**
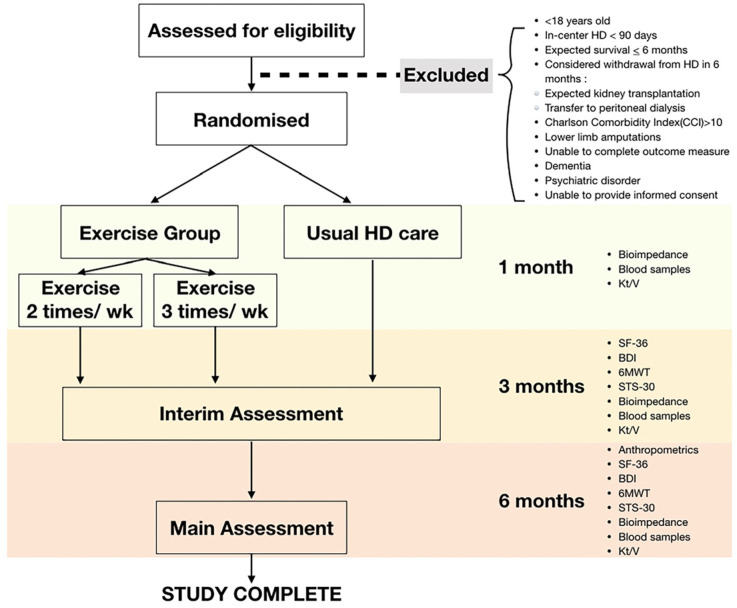
Study timeline.

## Data Availability

Not applicable.

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
