# Peer review of "Therapeutic Effects of Intradialytic Exercise on Life Quality of Patients with End-Stage Renal Disease: Study Protocol for a Randomized Control Trial"

_healthcare, 2022, doi:10.3390/healthcare10061103_

Round 1
Reviewer 1 Report
In this paper, the authors showed the study protocol for investigating the effects of an intradialytic exercise program delivered at different frequencies on HD patients in Taiwan. This paper should be of great interest to the researchers working in this field. However, several problems should be resolved for the manuscript to be accepted for publication in Healthcare.
Major points:
l This study protocol is for investigating the effect of intradialytic exercise; however, exercise habits other than the intradialytic period may affect results. Therefore, in the “Materials and Method” section, exercise habits at baseline and during the follow-up period should be investigated.
l In previous studies investigating the effects of exercise on CKD patients, follow-up periods varied from 12 weeks to 20 months (PMID 22113127). In this study protocol, the outcome is measured after the initiation of the study 6 months later. The authors should clarify the reason for this follow-up period in the “Material and Method” section.
Minor point:
l The authors highlighted that this study protocol is for investigating the effect of intradialytic exercise in Asian people. However, the “Introduction” section does not describe the information about the patient’s race in previous studies.
Overall, this is a well-written and interesting manuscript.
Author Response
Thanks for the reviewer’s detailed examination of our paper and the valuable comments, which help us to present our study more clearly. The manuscript has been revised based on these suggestions. Our responses to the reviewer’s comments are given below on a point-by-point basis.
- Thanks for the reviewer’s comment mentioning a very crucial potential confounder. We agree that exercise habits other than the intradialytic exercise program may affect results. Therefore, in the paragraph of 2.2.1 selection criteria, we added “Considering different baseline exercise habits may pose a significant confounding factor in this study, having no previous regular exercise habits is also listed to be one of our inclusion criteria.” Furthermore, we modified section 2.3.1 and 2.3.2 to add “The participants will be encouraged to maintain their regular physical activities as usual, and no extra exercise training should commence during the 6 months of following up period” in the “usual care” paragraph and “As in the usual care group, these participants are also encouraged to keep their original daily activities without additional exercise training during the study period” in the “intradialytic exercise” paragraph.
- We designed to measure our outcomes at 3 time points: right after the initiation of the study, 3 months and 6 months later according to previous studies on intradialytic exercise. We added the reason in the end of paragraph “2.1 study design overview” as following: “The timings of the outcome measurement are set according to past studies on intradialytic exercise. Some of them measured the results after 3 months of intradialytic exercise[13-15], while others evaluated 6 months later.[11,16]”
- In the introduction, we mainly mentioned the results from CYCLE-HD program of UK, and we’ve added the word “British” in the sentence “It also improves HD adequacy, has anti-depressive, cost-effective outcome, and better quality of life for these British patients[11,12]”
Reviewer 2 Report
Keywords: some terms don´t match with any subject in MeSH. Please, use indexed terms.
Introduction: line 43, there is only one reference to suppport thar low level of physical activity increases morbidity and mortality. Can you please provide more references to strengthen the comment.
In general, there are few references for such important statements in the background section.
Objective: it is not clear to me the reason of comparing two different frequencies of exercise routines. Given taht there is lack of evidence of the effects of intradialytic exercise programs, a comparison between usual care and one only frequency of exercise seems to be a more reasonable start in this research line.
Please, justify why one group with 3 sessions/week and another group with 2 sessions/week ¿Is there any evidence for the significance of the difference of frequency?

Author Response
Thanks for the reviewer’s detailed examination of our paper and the valuable comments, which help us to present our study more clearly. The manuscript has been revised based on these suggestions. Our responses to the reviewer’s comments are given below on a point-by-point basis.
- Thanks to the reviewer for the critical reminding. We’ve revised the key words to the MeSH indexed terms.
- We agree that to support the statement “low level of physical activity increases morbidity and mortality” by only one reference was too weak. Therefore, we found three additional systemic review articles to support this important statement and added it in our reference list.
- Although intradialytic exercise has not been routinely adopted in HD patients worldwide, there’re already some evidence to support the benefits of it, including the physical function performance, depressive status, and quality of life, as shown in our references 12-16. If we merely compare the therapeutic effects between usual care and one only frequency of exercise, the whole protocol will be very similar to the past studies, which may significantly reduce the novelty of this study.
- As we mentioned in the discussion section, most intradialytic exercises in the literature, including the best-known CYCLE-HD study, are conducted for 30 minutes duration and given 3 times a week. There is only one study that has used the exercise frequency of 2 times a week, but the training mode involved resistance training and stretching, not aerobic training [1]. Some other studies also adopted the exercise frequency of 2 to 3 times a week, such as those of Vilstreren et al., Chigira et al., and Bae et al. [2-4]. However, to our knowledge, no study has yet directly compared intradialytic exercise groups specifically between 3 times and 2 times a week. Since cycle ergometer devices and healthcare givers monitoring intradialytic exercise may be limited in some hospitals in different countries, whether the frequency of intradialytic exercise could be reduced is a crucial and practical issue that needs to be clarified. Hence, we have designed this comparative study in order to clarify this issue and try to fill the knowledge gap in the field of intradialytic exercise.
Reference:
- Chen JL, Godfrey S, Ng TT, Moorthi R, Liangos O, Ruthazer R, et al. Effect of intra-dialytic, low-intensity strength training on functional capacity in adult haemodialysis patients: a randomized pilot trial. Nephrology, dialysis, transplantation : official publication of the European Dialysis and Transplant Association - European Renal Association. 2010;25(6):1936-43.
- van Vilsteren MC, de Greef MH, Huisman RM. The effects of a low-to-moderate intensity pre-conditioning exercise programme linked with exercise counselling for sedentary haemodialysis patients in The Netherlands: results of a randomized clinical trial. Nephrology, dialysis, transplantation : official publication of the European Dialysis and Transplant Association - European Renal Association. 2005;20(1):141-6.
- Chigira Y, Oda T, Izumi M, Yoshimura T. Effects of exercise therapy during dialysis for elderly patients undergoing maintenance dialysis. Journal of physical therapy science. 2017;29(1):20-3.
- Bae YH, Lee SM, Jo JI. Aerobic training during hemodialysis improves body composition, muscle function, physical performance, and quality of life in chronic kidney disease patients. Journal of physical therapy science. 2015;27(5):1445-9.
Round 2
Reviewer 1 Report
The comments raised by the reviewer are well addressed.
Reviewer 2 Report
The manuscript is now suitable for publishing. The authors have reviewed and made the canges suggested by the reviewer satisfactorily.
